# Elastic Modulus and Elasticity Ratio of Malignant Breast Lesions with Shear Wave Ultrasound Elastography: Variations with Different Region of Interest and Lesion Size

**DOI:** 10.3390/diagnostics11061015

**Published:** 2021-06-01

**Authors:** Antonio Bulum, Gordana Ivanac, Eugen Divjak, Iva Biondić Špoljar, Martina Džoić Dominković, Kristina Bojanić, Marko Lucijanić, Boris Brkljačić

**Affiliations:** 1Department of Diagnostic and Interventional Radiology, University Hospital Dubrava, Avenija Gojka Šuška 6, 10000 Zagreb, Croatia; gordana.augustan@gmail.com (G.I.); edivjak@gmail.com (E.D.); iva.biondic2409@gmail.com (I.B.Š.); boris@brkljacic.com (B.B.); 2Department of Radiology, School of Medicine, University of Zagreb, Šalata 3, 10000 Zagreb, Croatia; 3Department of Radiology, General Hospital Orašje, 3rd Street, 76270 Orašje, Bosnia and Herzegovina; martina.dzoic@gmail.com; 4Department of Otorhinolaryngology, Neurosurgery and Radiology, Faculty of Dental Medicine and Health Osijek, J.J. Strossmayer University of Osijek, Crkvena ul. 21, 31000 Osijek, Croatia; Bojanic.kristina@gmail.com; 5Department of Biophysics and Radiology, Faculty of Medicine Osijek, J.J. Strossmayer University of Osijek, Ul. Josipa Huttlera 4, 31000 Osijek, Croatia; 6Department of Radiology, Health Center Osječko-Baranjska County, Ul. Josipa Huttlera 4, 31000 Osijek, Croatia; 7Hematology Department, University Hospital Dubrava, Avenija Gojka Šuška 6, 10000 Zagreb, Croatia; markolucijanic@yahoo.com

**Keywords:** breast, ultrasound, elastography, cancer, imaging

## Abstract

Shear wave elastography (SWE) is a type of ultrasound elastography with which the elastic properties of breast tissues can be quantitatively assessed. The purpose of this study was to determine the impact of different regions of interest (ROI) and lesion size on the performance of SWE in differentiating malignant breast lesions. The study included 150 female patients with histopathologically confirmed malignant breast lesions. Minimal (E_min_), mean (E_mean_), maximal (E_max_) elastic modulus and elasticity ratio (e-ratio) values were measured using a circular ROI size of 2, 4 and 6 mm diameters and the lesions were divided into large (diameter ≥ 15 mm) and small (diameter < 15 mm). Highest E_min_, E_mean_ and e-ratio values and lowest variability were observed when using the 2 mm ROI. E_max_ values did not differ between different ROI sizes. Larger lesions had significantly higher E_mean_ and E_max_ values, but there was no difference in e-ratio values between lesions of different sizes. In conclusion, when measuring the E_min_, E_mean_ and e-ratio of malignant breast lesions using SWE the smallest possible ROI size should be used regardless of lesion size. ROI size has no impact on E_max_ values while lesion size has no impact on e-ratio values.

## 1. Introduction

Ultrasound elastography is an ultrasound technology which detects malignant breast lesions by measuring the changes of the elastic properties of breast tissue. There are two types of ultrasound elastography, strain and shear wave elastography (SWE) [1,2,3,4].

Strain elastography is performed by manual compression using the transducer which then produces an image based on the resulting displacement of the breast tissue caused by the compression. However, it is difficult to measure the exact amount of the applied force during compression resulting in the method being difficult to standardize. Additionally, the absolute elasticity values cannot be calculated, only qualitative results can be obtained [5]. Unlike strain elastography, SWE is a type of ultrasound elastography where the elastic properties of breast tissue can be both qualitatively and quantitatively assessed. The quantitative results are obtained by measuring the Young’s modulus of tissue elasticity in kilopascals (kPa), based on the focused ultrasound beam inducing mechanical vibrations and tissue displacement resulting in shear waves perpendicular to ultrasound beam direction. The velocity of shear waves can be measured and is proportional to tissue elasticity, allowing elastic modulus to be calculated. These values can be color-coded in various colors ranging between red and dark blue, which are superimposed over the gray-scale image in real time (so-called elastogram) [6]. In addition to the elastic modulus values, the elasticity ratio (e-ratio) can also be measured and provides another quantitative tool for the assessment of the tissue in focus [7]. The SWE acquisition box can be enlarged and positioned over the examined parts of the breast, encompassing focal lesions and surrounding fatty tissues. The elastic modulus is measured using a region of interest (ROI) circle that is positioned over the examined lesion in the breast, and a second ROI is positioned over the surrounding fatty tissue to calculate the e-ratio value between the two. The influence of the size of the ROI on the measured values of the elasticity modulus was addressed in only a few other publications assessing the utility of SWE in diagnosing breast cancer. In addition to that, the studies differed in methodology in regard to the ROI size, from using a ROI size of 1 mm to encompassing the entire lesion [8,9,10].

The aim of this study was to investigate the relation of the ROI area and the size of the breast lesion and their influence on the measured values of minimal (E_min_), mean (E_mean_), maximal elastic modulus (E_max_) and e-ratio of malignant breast lesions.

## 2. Materials and Methods

This retrospective single-center study conducted at our institution included 150 female patients (ages 37–92, median age 59) with histopathologically confirmed malignant breast lesions (PHD: invasive ductal carcinoma) between September 2017 and February 2019. Patients were excluded from the study if they had previous breast cancer, underwent surgery of the breast, chemotherapy, hormonal or radiation therapy. All patients were examined using a 4–15 MHz frequency 40 mm long linear probe, on the same ultrasound scanner (Aixplorer^®^, SuperSonic Imagine, Aix, France, product version 6.2.0, software 6.2.23751). Grey scale and shear-wave sonoelastography examinations were performed by an experienced breast radiologist with over 15 years of practice in breast ultrasound. The regions of interest used for the elastography measurements were set at diameters of 2 mm, 4 mm, and 6 mm. The stiffest areas of the lesion in the SWE acquisition box, including the immediately adjacent stiff tissue, were measured at the center of the ROI while the corresponding e-ratio value was obtained by placing a second ROI over the surrounding fatty tissue (Figure 1, Figure 2 and Figure 3) [11,12,13].

Additionally, all lesions were divided into small (diameter < 15 mm, Nm = 95) and large lesions (diameter ≥ 15 mm, Nm = 55) [8]. Normality of distribution of numerical variables was assessed using the Shapiro–Wilk test. Since most measurements were non-normally distributed, we presented all numerical variables as median and interquartile range (IQR) and used non-parametrical statistical tests. Coefficient of variation was obtained by dividing standard deviation of multiple assessments with the mean value. Measurements were mutually compared using the Friedman ANOVA for repeated measures and between different lesion sizes using the Mann Whitney U test. Correlation between different measurements was assessed using the Spearman rank correlation. *p* values < 0.05 were considered to be statistically significant. Bonferroni correction for multiple comparisons was used where appropriate (<0.017 considered to be statistically significant). All analyses were done using the MedCalc statistical software ver. 19.7.

## 3. Results

SWE measures in a total of 150 patients were analyzed. There was a significant difference between SWE parameters, with E_mean_, standard deviation, coefficient of variation, E_min_ and e-ratio values all being statistically significantly different between different ROI sizes (*p* < 0.017; Table 1; Figure 4).

Highest E_min_, E_mean_ and e-ratio values and lowest variability were observed when using 2 mm in comparison to other ROI sizes. It should be noted that the lowest detected e-ratio values were 3.7, 2 and 1.3 for 2 mm, 4 mm, and 6 mm ROI sizes, respectively, showing that using higher ROI values results in substantially lower e-ratio measurements. However, E_max_ values did not differ according to different ROI sizes. Regarding lesion size, larger lesions had significantly higher E_mean_ and E_max_ values as assessed using all three ROI sizes (*p* < 0.05; Table 2; Figure 5).

However, significant differences between lesion size and other SWE parameters like e-ratio (significant difference for 2 mm ROI size) and E_min_ values (significant differences for 2 mm and 4 mm ROI sizes) were present only when lower ROI sizes were used whereas variability of measurement did not differ according to the lesion size (Table 2). Median absolute differences between ROI measurements were 26 kPa for 2 mm vs. 4 mm, 23 kPa for 4 mm vs. 6 mm, and 50 kPa for 2 mm vs. 6 mm. Differences between 2 and 4 mm and 2 and 6 mm ROI were statistically significantly more pronounced with larger lesion size (*p* < 0.05; Table 2) and with higher e-ratio values (*p* < 0.05, Table 3) suggesting that with larger lesions it is more likely to obtain different measurements when using different ROI sizes.

## 4. Discussion

The influence of the ROI size on the measured values of the E_min_, E_mean_ and e-ratio of malignant breast lesions is significant. The highest values and lowest variability were observed when using the 2 mm diameter in comparison to other ROI sizes. The E_mean_ value is calculated as a sum of elasticities of all pixels within the ROI, divided by the number of pixels. Therefore, the value of E_mean_ is influenced if the lower elasticity values from the tissues surrounding the lesion are included into the measurements [7]. When a larger ROI is used to measure the lower elasticity values of the surrounding tissues that are not influenced by the malignant lesion, the effect on the measured E_min_ and e-ratio is the same as on the measured E_mean_ values. E_max_ values, on the other hand, are not influenced by the change in the ROI size, and it was demonstrated in multiple studies that significant differences in its values were present between benign and malignant breast lesions versus normal breast tissues [8,14,15,16,17]. Regarding the lesion size, larger lesions had significantly higher E_mean_ and E_max_ values as assessed using all three ROI sizes which is in line with the results from previous studies where some of these values were assessed in malignant breast lesions [8,18]. However, there was no significant change in e-ratio values between small and large lesions when a 2 mm ROI was used.

In conclusion, when measuring the E_min_, E_mean_ and e-ratio of malignant breast lesions using SWE the smallest possible ROI size should be used regardless of lesion size. The effect of the ROI size is most pronounced on the E_mean_ values and can influence the measured elasticity values and, subsequently, sensitivity and specificity of the examination. If different ROI sizes are used, E_max_ values can be measured but different values will be obtained depending on the size of the lesion. In that case, the use of e-ratio with the smallest possible ROI is recommended to achieve uniform results regardless of lesion size. There are some limitations to this retrospective, single-center study. All lesions in this study were histologically confirmed as malignant, therefore the performance of SWE to differentiate between benign and malignant breast lesions was not assessed. The lesions were also divided into groups of small and large lesions by consideration of the mean value of their diameter (15 mm). Hence, further research defining the optimal cut-off thresholds of the lesion diameters is needed. Additionally, the influence of lesion depth and breast thickness was not considered in this study and it is suggested that both might influence the performance of SWE [19]. It is also known that soft tissues are non-linear and that any pressure applied by the probe may influence the results. In this study, minimal manual compression of the examined region was applied to avoid false positive results, as suggested by previous research [20]. A recent study also suggested that some amount of compression is essential to avoid false negative results [21]. However, these are also areas that also require further research.

## Figures and Tables

**Figure 1 diagnostics-11-01015-f001:**
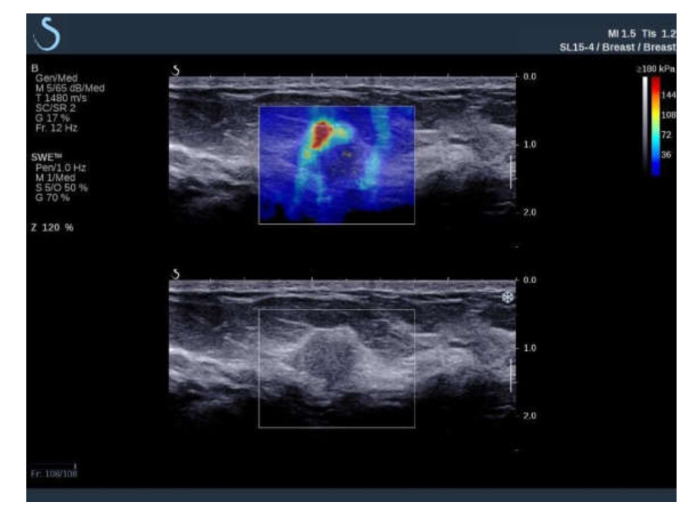
Ultrasound examination of a malignant breast lesion with B-mode (**bottom**) and shear wave elastography (**top**) where elastic properties of breast tissues are displayed qualitatively, color-coded, and superimposed on the B-mode image. Red color denotes areas with the highest elastic modulus values.

**Figure 2 diagnostics-11-01015-f002:**
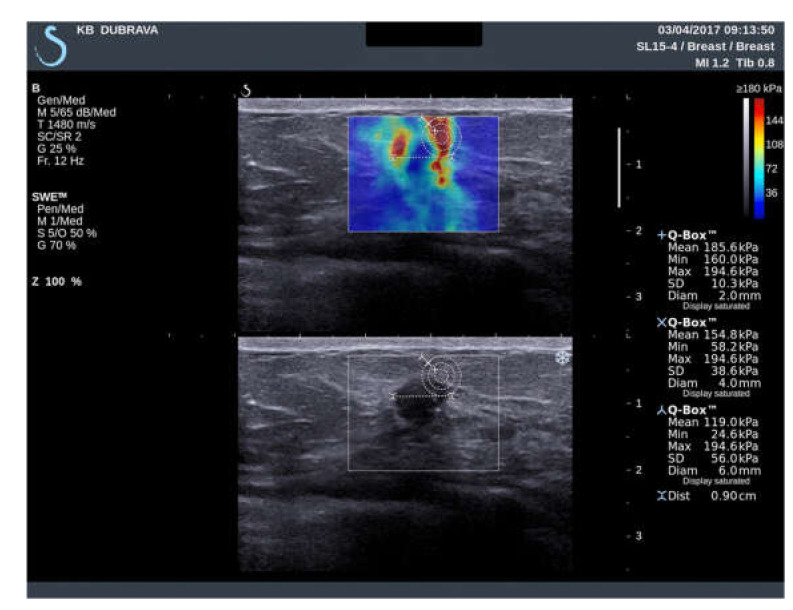
Ultrasound examination with shear wave elastography of a hypoechoic breast lesion, later pathohistologically confirmed after core-biopsy as an invasive ductal carcinoma. Elastic modulus values measurements with ROI size of 2 mm, 4 mm and 6 mm were performed and the measured values are displayed.

**Figure 3 diagnostics-11-01015-f003:**
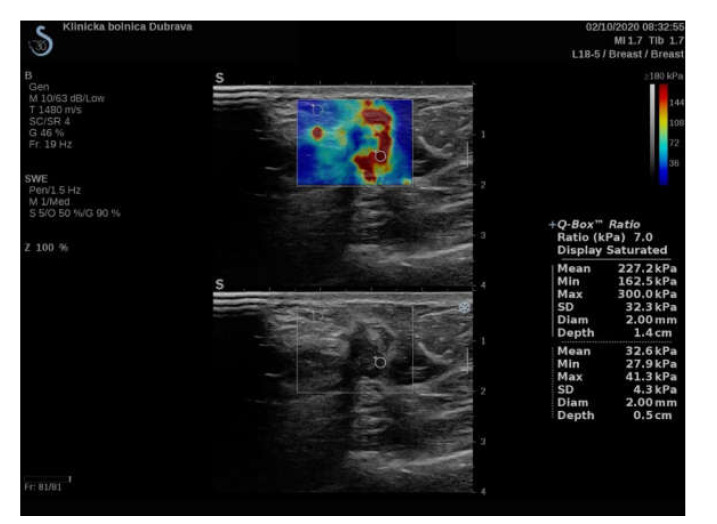
Ultrasound examination with shear wave elastography of a hypoechoic breast lesion, later pathohistologically confirmed after core-biopsy as an invasive ductal carcinoma. An e-ratio value measurement with ROI size of 2 mm with the measured values is displayed (measured mean values are used to calculate the ratio). Additional ROI with a size of 4 mm and 6 mm were placed at the same points as the 2 mm ROI to obtain additional values (not displayed in this image).

**Figure 4 diagnostics-11-01015-f004:**
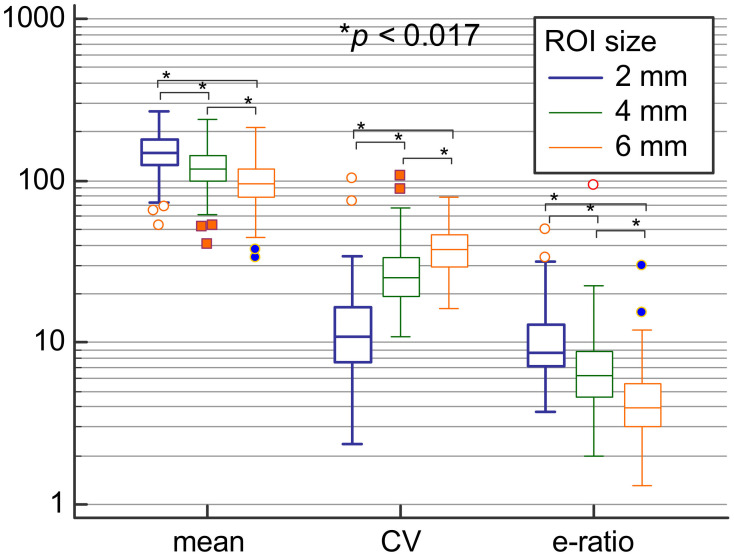
Comparison of means, coefficients of variation and e-ratio values between 2 mm, 4 mm, and 6 mm ROI sizes.

**Figure 5 diagnostics-11-01015-f005:**
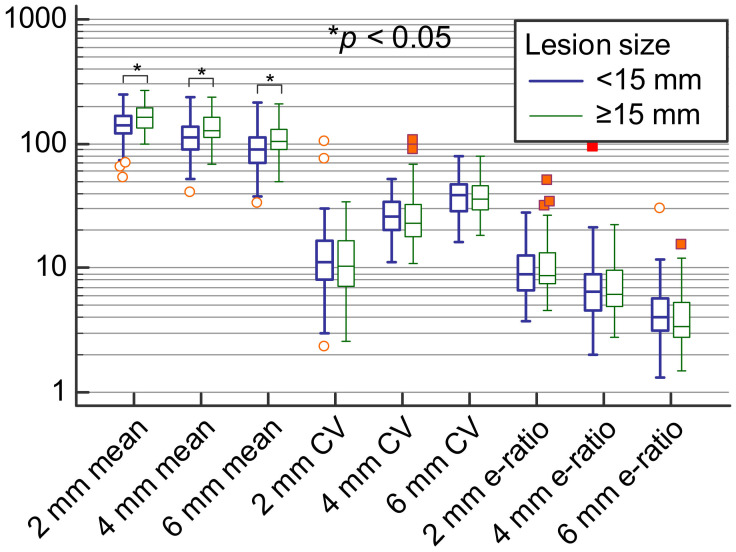
Means, coefficients of variation and e-ratio measurements for ROI 2 mm, 4 mm and 6 mm sizes stratified according to the lesion size.

**Table 1 diagnostics-11-01015-t001:** Pairwise comparisons of different SWE measures according to the ROI size.

Variable	ROI 2 mm (1)	ROI 4 mm (2)	ROI 6 mm (3)	*p* Value
E_mean_ (kPa)	148.85	117.55	95.65	**<0.001 * (1 vs. 2; 1 vs. 3; 2 vs. 3) ***
Sd (kPa)	16.6	31.15	36.65	**<0.001 * (1 vs. 2; 1 vs. 3; 2 vs. 3) ***
Coeff. var. (%)	10.9	25.22	37.77	**<0.001 * (1 vs. 2; 1 vs. 3; 2 vs. 3) ***
E_min_ (kPa)	108.35	50.8	23.85	**<0.001 * (1 vs. 2; 1 vs. 3; 2 vs. 3) ***
E_max_ (kPa)	184.45	184.45	184.45	0.236
e-ratio	8.7	6.3	3.95	**<0.001 * (1 vs. 2; 1 vs. 3; 2 vs. 3) ***

* Statistically significant at level *p* < 0.05 (Bonferroni corrected *p* < 0.017 for three simultaneous pairwise comparisons). Results are presented as median values. Friedman ANOVA for repeated measures was used. Abbreviations: ROI—region of interest; sd—standard deviation; coeff. var.—coefficient of variation; min—minimum; max—maximum.

**Table 2 diagnostics-11-01015-t002:** Different SWE measurements stratified according to the lesion size.

Variable	Lesion Size < 15 mm	Lesion Size ≥ 15 mm	*p* Value	Correlation with Lesion Size as a Cont. Variable
Nm of patients	95	55	-	-
E_mean_ ROI 2 mm (kPa)	141.8 IQR (119.6–165.9)	161.7 IQR (134.95–190.6)	**0.003 ***	Rho = 0.4; ***p* < 0.001 ***
Sd ROI 2 mm (kPa)	16.6 IQR (11–23)	16.6 IQR (11.35–23.3)	0.729	Rho = 0.18; ***p* = 0.030 ***
Coeff. var. ROI 2 mm (%)	11.1 IQR (8.08–16.42)	10.1 IQR (6.7–16.32)	0.245	Rho = −0.01; *p* = 0.894
E_min_ ROI 2 mm (kPa)	104.9 IQR (79.4–127.4)	123.8 IQR (91.3–154.6)	**0.015 ***	Rho = 0.3; ***p* < 0.001 ***
E_max_ ROI 2 mm (kPa)	171.8 IQR (137.9–192)	192 IQR (171.85–221.6)	**0.001 ***	Rho = 0.44; ***p* < 0.001 ***
e-ratio ROI 2 mm	8.8 IQR (6.6–12.75)	8.6 IQR (7.5–12.95)	0.201	Rho = 0.19; ***p* = 0.023 ***
E_mean_ ROI 4 mm (kPa)	111.3 IQR (89.15–135.55)	128.6 IQR (111.9–161.8)	**0.001 ***	Rho = 0.42; ***p* < 0.001 ***
Sd ROI 4 mm (kPa)	31 IQR (21.9–37.65)	32.8 IQR (24.15–40.1)	0.256	Rho = 0.21; ***p* = 0.009 ***
Coeff. var. ROI 4 mm (%)	25.7 IQR (20.72–34.14)	22.7 IQR (17.93–32.1)	0.154	Rho = −0.1; *p* = 0.219
E_min_ ROI 4 mm (kPa)	49.3 IQR (32.3–66.9)	57.2 IQR (40.05–82)	0.063	Rho = 0.26; ***p* = 0.002 ***
E_max_ ROI 4 mm (kPa)	171.8 IQR (137.9–192)	192 IQR (171.85–214.6)	**0.001 ***	Rho = 0.43; ***p* < 0.001 ***
e-ratio ROI 4 mm	6.3 IQR (4.55–8.8)	6.3 IQR (4.95–10.05)	0.638	Rho = 0.14; *p* = 0.092
E_mean_ ROI 6 mm (kPa)	89.4 IQR (69.8–110.1)	105.1 IQR (89.65–129.8)	**0.001 ***	Rho = 0.4; ***p* < 0.001 ***
Sd ROI6 mm (kPa)	34.5 IQR (24.75–43.45)	39.7 IQR (32.95–46.55)	**0.005 ***	Rho = 0.39; ***p* < 0.001 ***
Coeff. var. ROI6 (kPa)	38.4 IQR (28.82–46.59)	35.7 IQR (29.24–44.9)	0.785	Rho = 0.01; *p* = 0.942
E_min_ ROI 6 mm (kPa)	23.1 IQR (15.1–38.45)	25.4 IQR (10.85–41.1)	0.685	Rho = 0.09; *p* = 0.295
E_max_ ROI 6 mm (kPa)	171.8 IQR (137.9–192)	192 IQR (171.85–221.6)	**0.001 ***	Rho = 0.43; ***p* < 0.001 ***
e-ratio ROI 6 mm	4 IQR (3.15–5.7)	3.4 IQR (2.85–5.35)	0.335	Rho = −0.05; *p* = 0.527
ROI 2 mm vs. 4 mm mean difference	26.3 IQR (18.55–37.95)	26.2 IQR (21.4–38.75)	0.408	Rho = 0.13; *p* = 0.116
ROI 2 mm vs. 6 mm mean difference	47.4 IQR (34.05–64.6)	52.7 IQR (41.45–69)	0.116	Rho = 0.22; ***p* = 0.006 ***
ROI 4 mm vs. 6 mm mean difference	22.2 IQR (15.65–26)	26.1 IQR (18.25–32.2)	**0.008 ***	Rho = 0.32; ***p* < 0.001 ***

* Statistically significant at level *p* < 0.05. Results are presented as median values and interquartile range. Mann–Whitney U test and Spearman rank correlation were used. Abbreviations: ROI—region of interest; cont.—continuous; sd—standard deviation; coeff. var.—coefficient of variation; min—minimum; max—maximum.

**Table 3 diagnostics-11-01015-t003:** Correlation between e-ratio assessed by different ROI sizes and differences in SWE measurements.

Variable	e-Ratio ROI 2 mm	e-Ratio ROI 4 mm	e-Ratio ROI 6 mm
ROI 2 mm vs. 4 mm mean difference	Rho = 0.14; *p* = 0.088	Rho = 0.07; *p* = 0.388	Rho = −0.01; *p* = 0.952
ROI 2 mm vs. 6 mm mean difference	Rho = 0.25; ***p* = 0.002 ***	Rho = 0.2; ***p* = 0.017 ***	Rho = 0.08; *p* = 0.362
ROI 4 mm vs. 6 mm mean difference	Rho = 0.39; ***p* < 0.001 ***	Rho = 0.39; ***p* < 0.001 ***	Rho = 0.22; ***p* = 0.007 ***

* Statistically significant at level *p* < 0.05/Spearman rank correlation was used. Abbreviations: ROI—region of interest.

## Data Availability

Data available on request due to restrictions. The data presented in this study is available on request from the corresponding author. The data is not publicly available due to patient privacy reasons.

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
