# Peer review of "Elastic Modulus and Elasticity Ratio of Malignant Breast Lesions with Shear Wave Ultrasound Elastography: Variations with Different Region of Interest and Lesion Size"

_diagnostics, 2021, doi:10.3390/diagnostics11061015_

Round 1

Reviewer 1 Report

The manuscript presents a statistical analysis of the variation on the elastic modulus and elastic ratio of breast cancer regarding the size of the ROI by using SW elastography. The study comprises a reasonably large number of cases, nevertheless, the manuscript seems too simple for a work of eight authors. Most importantly, the paper seems to be merely incremental. The original contribution of the paper concerning previous studies has not been clearly stated. The authors are suggested to generally improve the manuscript:

  • The readers would benefit from an explanation that supports the expectation of obtaining different results depending on the ROI size.
  • The authors have forgotten to state the original contribution of the manuscript in comparison to the few articles that have analysed the effect of the ROI size on the results.
  • The authors are suggested to further elaborate the methodology section by adding detail on settings of the SSI system for performing elastography. Some mention of the basic physics principle of SWE would also be appreciated by readers with little fundamental knowledge of dynamic elastography.
  • Lines 78-83: please rephrase for better reading and understanding of the sentence.
  • The authors are encouraged to briefly discuss the influence of the pressure exerted on the tissue by the probe on the results. It is well known that soft tissue is non-linear and that the shear wave velocity increases when the tissue is under pressure.

Reviewer 2 Report

In the manuscript, the influence of the region of interest and lesion sizes on breast shear wave elastography performance is investigated. Aixplorer, SuperSonic Imagine ultrasound machine was used. The study included 150 female patients with histopathologically confirmed malignant breast lesions. Based on the results,  smallest possible ROI is recommended.

Page 1: Abstract:  “Emax values did not significantly differ between different ROI sizes.” Actually, based on the presented results, there is no difference between Emax values at all.

Page 2: “…while the corresponding e-ratio value was obtained by placing a second ROI 82 over the surrounding fatty tissue…” Is e-ration defined as a ration between mean values?

Page 8: “there was no significant change in e-ratio values between small and large lesions.” This result is required additional discussion. If mean value of Young’s modulus for larger tumors is higher, why the ratio to fat does not depend on the size? Does it mean that fat tissue is also stiffer for larger tumors?

Round 2

Reviewer 1 Report

Thank you for your correct responses and modifications. The paper has improved.